

# Identification of *YUC* genes associated with leaf wrinkling trait in Tacai variety of Chinese cabbage

Xuelian Ye[1], Ji Sun[2], Yuan Tian[1], Jingwen Chen[1], Xiangtan Yao[3], Xinhua Quan[3] and Li Huang[1]

[1] College of Agriculture and Biotechnology, Zhejiang University, Hangzhou, China
[2] College of Agriculture and Biotechnology, Wenzhou Vocational College of Science and Technology (Wenzhou Academy of Agricultural Sciences), Wenzhou, China
[3] Jiaxing Academy of Agricultural Sciences, Jiaxing, China

## ABSTRACT

Chinese cabbage (*Brassica campestris* L. ssp. *chinensis* (L.) Makino) stands as a widely cultivated leafy vegetable in China, with its leaf morphology significantly influencing both quality and yield. Despite its agricultural importance, the precise mechanisms governing leaf wrinkling development remain elusive. This investigation focuses on 'Wutacai', a representative cultivar of the Tacai variety (*Brassica campestris* L. ssp. *chinensis* var. *rosularis* Tsen et Lee), renowned for its distinct leaf wrinkling characteristics. Within the genome of 'Wutacai', we identified a total of 18 *YUCs*, designated as *BraWTC_YUCs*, revealing their conservation within the *Brassica* genus, and their close homology to *YUCs* in Arabidopsis. Expression profiling unveiled that *BraWTC_YUCs* in Chinese Cabbage exhibited organ-specific and leaf position-dependent variation. Additionally, transcriptome sequencing data from the flat leaf cultivar 'Suzhouqing' and the wrinkled leaf cultivar 'Wutacai' revealed differentially expressed genes (DEGs) related to auxin during the early phases of leaf development, particularly the *YUC* gene. In summary, this study successfully identified the *YUC* gene family in 'Wutacai' and elucidated its potential function in leaf wrinkling trait, to provide valuable insights into the prospective molecular mechanisms that regulate leaf wrinkling in Chinese cabbage.

## INTRODUCTION

Chinese cabbage (*Brassica campestris* L. ssp. *chinensis* (L.) Makino), a member of the Cruciferae family, is indigenous to China and extensively cultivated across China and East Asia. As a prominent leafy vegetable crop, the attributes of its leaves hold significant importance for its economic value and agricultural breeding strategies. Chinese cabbage exhibits a diverse range of leaf traits, encompassing leaf shape, margin, and surface features. These variations not only profoundly influence the quality, yield, and resistance, but also serve as morphological indicators for the classification of different subspecies and varieties within the species. The "Specification for Description of non-heading Chinese cabbage Germplasm Resources" classifies leaf wrinkling into four categories: flat, slightly

Corresponding author
Li Huang, lihuang@zju.edu.cn

wrinkled, wrinkled, and highly wrinkled. The cultivar 'Wutacai' which belongs to the Tacai variety (var. *rosularis* Tsen et Lee), exhibits obvious leaf wrinkling characterized by regular ridges ascending towards the leaf axis, and is thus classified as highly wrinkled. Investigation into the leaf wrinkling of Tacai offers the potential insights not only into the leaf wrinkling of Chinese cabbage but also into the molecular mechanisms regulating leaf development.

However, the mechanisms underlying the formation and regulation of leaf wrinkling characteristics in plants remain poorly understood. Previous studies have identified a wrinkled leaf mutant, *cin*, in *Kalanchoe daigremontiana*, attributed to a mutation in a TCP (TEOSINTE BRANCHED1/CYCLOIDEA/PROLIFERATING CELL FACTORS) protein, with the *cin* mutation leading to excessive growth in the leaf margin region, leading to leaf wrinkling (Nath et al., 2003). TCP4, another member of the TCP family, has also been implicated in leaf development. Transgenic Arabidopsis (*Arabidopsis thaliana*) plants overexpressing *TCP4::7mTCP4* exhibited upward bending of leaves. Furthermore, a miRNA-targeted gene loss-of-function mutant, *phb-6* (PHABULOSA, a member of the Class III HD-Zip family), displayed subtle bending phenotypes in leaves (Liu et al., 2010). The *GhBOP1* (Block of proliferation 1) gene, crucial in ribosome maturation and cell cycle progression, when silenced in upland cotton (*Gossypium hirsutum*) plants, resulted in the wrinkling phenotype of young leaves (Wang et al., 2022b). In Chinese cabbage, a stably inherited wrinkled leaf mutant named *lcm* was identified through EMS mutagenesis, with the candidate gene *BraA01g007510.3C* found to encode an H$^+$ATPase 2 (Zhang, 2020). Functional investigations of these genes have provided valuable insights into the formation of leaf wrinkling characteristics. However, while leaf wrinkling characteristics in Chinese cabbage share similarities with those observed in the aforementioned mutants, they also exhibit distinct features. Unlike the partial bulges seen in those mutants, Chinese cabbage leaf wrinkling, particular in cultivar 'Wutacai', is characterized by highly wrinkled leaves, a trait that is heritable. The underlying mechanisms governing the formation and regulation of leaf wrinkling trait in Chinese cabbage necessitate further investigation.

Auxin, as a pivotal endogenous hormone, exerts a profound influence on plant growth and development. The TRYPTOPHAN AMINOTRANSFERASE OF ARABIDOPSIS (TAA)/YUCCA (YUC) pathway is widely recognized as the most critical and well-defined pathway for auxin production (Korasick, Enders & Strader, 2013). Within this pathway, YUC proteins (YUCs) function as flavin monooxygenases (FMOs), catalyzing the rate-limiting and irreversible oxidation decarboxylation of indole-3-pyruvic acid (IPyA) to yield indole-3-acetic acid (IAA). YUCs represent the earliest-discovered FMO proteins in plants, exhibiting structural and functional properties similar to mammalian FMOs (Cashman, 2002). Both bioinformatic and biochemical studies have revealed that plant FMO proteins typically contain six highly conserved motifs (Schlaich, 2007). Particularly noteworthy are the flavin adenine dinucleotide (FAD)-binding motif and the nicotinamide adenine dinucleotide phosphate (NADPH)-binding motif, which harbor highly conserved GxGxxG sequences recognized as integral components of the classical Rossman fold. Mutations affecting these motifs in maize (*Zea mays*) (Gallavotti et al., 2008) and Arabidopsis (Schlaich, 2007) have led to the complete loss of *YUC* function, highlighting

their crucial role as essential conserved motifs for YUC protein functionality (*Cao et al., 2019*).

The *YUC* genes (*YUCs*) were initially identified in Arabidopsis *via* activation tagging lines, which manifest developmental anomalies, such as downward curled leaves and semi-erect growth, phenotypic characteristics attributed to an excess of auxin (*Zhao et al., 2001*). Compared to proteins involved in auxin transport and signal transduction, the gene family encoding YUCs is comparatively small. In recent years, comprehensive whole-genome phylogenetic analyses have unveiled the presence of the *YUC* gene family in over 20 plant species, encompassing Arabidopsis, rice (*Oryza sativa*), maize, cucumber (*Cucumis sativus*), and wild strawberry (*Fragaria vesca*), with 11, 14, 14, 10, and 8 family members, respectively (*Cao et al., 2019*).

The local concentration of auxin intricately governs the normal morphogenesis of plant leaves. For instance, the polar differentiation of leaf primordia, delineating the proximal-distal axis, is orchestrated by the localized accumulation of auxin in leaf margin cells, a process modulated by multiple YUCs (*Zgurski et al., 2005*; *Wang et al., 2011*). Overexpression of *FaYUC1* in Arabidopsis results in narrow and downward curled leaves (*Liu et al., 2012*). Mutations in *YUC1*, *YUC2*, *YUC4* and *YUC6* lead to diminished vein and vascular bundle numbers, with the severity of the phenotype being depended on the dosage of these four *YUCs* (*Cheng, Dai & Zhao, 2006*). Transgenic potato (*Solanum tuberosum* L.) overexpressing *AtYUC6* displays a high-auxin phenotype, characterized by narrow, downward-curled leaves and increased plant height (*Kim et al., 2013*). The rapeseed (*Brassica napus* L.) *ed1* mutant plants exhibit dwarfing and leaf wrinkling, a phenotype similar to that of some Arabidopsis mutants deficient in auxin biosynthesis. Map-based cloning result displayed that *ED1* encodes a protein homologous to *AtIAA7*. Meanwhile, in *ProED1::ed1* and *35S::ed1* plants, the expression of several genes related to IAA synthesis is up-regulated (*Zheng et al., 2019*). These findings collectively underscore the involvement of *YUCs* in plant leaf morphogenesis, where their overexpression, along with an elevation in local auxin concentration in Arabidopsis and other plant species, frequently results in leaf curling. Leaf curling and wrinkling both exhibit uneven leaf surfaces, with the former being predominantly localized, while the latter features more regular and extensive bugles. It is intriguing to explore whether the local concentration of auxin is involved in the formation of leaf wrinkling in Chinese cabbage.

This study aimed to investigate the potential involvement of the *YUC* gene family in regulating the leaf wrinkling trait in Chinese cabbage. Utilizing bioinformatics methodologies, 18 *YUCs* were identified based on the genome data of the Chinese cabbage variety, 'Wutacai', renowned for its distinctive leaf wrinkling trait. Subsequent structural and phylogenetic analyses of the *YUC* gene family in Chinese cabbage were conducted. Through gene expression data sourced from various organs available in the *Brassica* database BRAD, we pinpointed the organ-specific expression of *YUCs* in Chinese cabbage. Transcriptome sequencing data obtained from the 1st and the 5th leaf of 'Wutacai' and 'Suzhouqing' showed that *YUC* may be involved in the formation of leaf wrinkling trait. Additionally, we validated and compared the expression patterns of the predominant *BrYUC3* and *BrYUC6* across leaves in different positions, employing 'Wutacai' and the

flat-leaved representative variety 'Youqing 49' through RT-qPCR. This investigation seeks to elucidate the potential association of the *YUC* gene family with the leaf wrinkling trait, thereby laying the groundwork for molecular breeding in Chinese cabbage and charting future research avenues pertaining to plant leaf development.

## MATERIALS AND METHODS

### Identification of *YUCs* in 'Wutacai'

The protein sequences of *YUCs* in 'Wutacai' were obtained from a previous investigation (*Cai et al., 2021*). Subsequently, YUC protein sequences for cucumber, maize, Arabidopsis, and rice were obtained by querying the National Center for Biotechnology Information (NCBI) website using the keyword "YUC". These sequences were then individually aligned with the protein sequences from 'Wutacai' using TBtools (*Chen et al., 2023*) to conduct pairwise sequence alignment. The intersection of these four alignment results produced a set of 40 sequences. Following this, the 40 protein sequences from 'Wutacai' were subjected to the NCBI Conserved Domain Database (CDD) online platform (http://www.ncbi.nlm.nih.gov/Structure/cdd/cdd.shtml) for gene structure domain prediction. Additionally, YUC protein sequence files of *Brassica oleracea* and *Brassica nigra* were acquired from the BRAD website (http://brassicadb.cn/#/Transcriptome/) utilizing the "Search-Syntenic Gene @ Subgenomes" function. This process involved inputting the gene ID of Arabidopsis to identify syntenic genes in those genomes. Specifically, the genome "Braol_JZS_V2.0 *Brassica oleracea* cv. JZS V2.0" was utilized for *B. oleracea*, while the genome "Brani_Ni100_V2 *Brassica nigra* cv. Ni100. V2.0" was employed for *B. nigra*.

### Construction of the phylogenetic tree

In order to explore the evolutionary relationships of the *YUCs* in 'Wutacai', all identified YUC protein amino acid sequences from 'Wutacai', *B. oleracea*, *B. nigra*, Arabidopsis, cucumber, rice, and maize were aligned using the "One Step Build a ML Tree" feature within the TBtools software. The maximum likelihood (ML) method was utilized to construct the phylogenetic tree with default parameters.

### Bioinformatic analysis of protein sequences

The protein sequences of *YUCs* in 'Wutacai' were subjected to analysis using online tools to assess protein physicochemical properties and predict subcellular localization. Specifically, we utilized the ProtParam (https://web.expasy.org/protparam/) and PSORT (https://wollpsort.hgc.jp/) tools for this purpose.

### Conservation motif and gene structure analysis

The Simple MEME Wrapper tool within the TBtools software was employed with default parameters to identify the conserved motifs within the 'Wutacai' YUC protein sequences. Additionally, gene structure domains were predicted using the online analysis website NCBI CDD (http://www.ncbi.nlm.nih.gov/Structure/cdd/cdd.shtml). Finally, TBtools was utilized to visualize both the conserved motifs and gene structure patterns.

## Prediction of *cis*-acting elements of promoters

The upstream 2,000 bp sequences of the coding regions of the 'Wutacai' *YUCs* were extracted using the Gtf/Gff3 sequences Extract tool in TBtools. Then the data were submitted to PlantCare (http://bioinformatics.psb.ugent.be/webtools/plantcare/html/) for predicting *cis*-acting elements. The results were visualized using the BioSequence viewer tool in TBtools.

## Expression analysis

The expression data of *YUCs* in various organs of Chinese cabbage were acquired from the BRAD database, specifically from the "GSE43245" library. Transcriptome sequencing was conducted on two early stages of leaf morphogenesis in the 1st and 5th leaves of 'Wutacai' and 'Suzhouqing'. The raw sequence data reported in this article have been deposited in the Genome Sequence Archive (*Chen et al., 2021*) in National Genomics Data Center (*CNCB-NGDC Members & Partners, 2022*), Beijing Institute of Genomics, Chinese Academy of Sciences (GSA: CRA015400 and CRA015503) that are publicly accessible at https://ngdc.cncb.ac.cn/gsa (*Chen et al., 2021*; *CNCB-NGDC Members and Partners, 2022*). Visualization of the data was achieved using the HeatMap tool within TBtools.

## RT-qPCR experiment

Total RNA was extracted by Trizol Reagent according to the manufacturer's instructions (Invitrogen, Carlsbad, CA, USA) and reverse transcribed into first strand cDNA using HiScript II RT SuperMix for qPCR (+gDNA wiper) (Vazyme, Nanjing, China). With the reference gene *BrUBC10*, RT-qPCR reactions were performed using the SYBR Green Premix Pro Taq HS qPCR Kit (AG, Changsha, China) in a CFX96 Real-Time System (Bio-Rad, California, USA). Primers used in the experiment are provided in Table S1. The results were calculated using the $2^{-\Delta\Delta Ct}$ method (*Livak & Schmittgen, 2001*).

## RESULTS

### Identification and phylogenetic analysis of the *YUC* gene family

A total of 18 *YUCs* named *BraWTC_YUCs* were identified in the 'Wutacai' genome, while *B. oleracea*, *B. nigra*, Arabidopsis, cucumber, rice, and maize genomes were determined to possess 18, 18, 11, 10, 14 and 14 *YUCs* respectively. To investigate the evolutionary relationships within the *YUC* gene family, phylogenetic trees were constructed utilizing the protein sequences of the *YUCs* in these species (Fig. 1A). The results revealed that the 'Wutacai' *YUCs* were most closely related to those of *B. oleracea* and *B. nigra*, both of which belong to the same genus as 'Wutacai'. Secondary to this relationship, a close relationship was observed with those of Arabidopsis. However, the evolutionary relationship between 'Wutacai' and rice was notably distinct, as was the case with 'Wutacai' and maize. This phenomenon may be attributed to the monocotyledonous nature of maize and rice, while 'Wutacai' is a dicotyledonous plant.
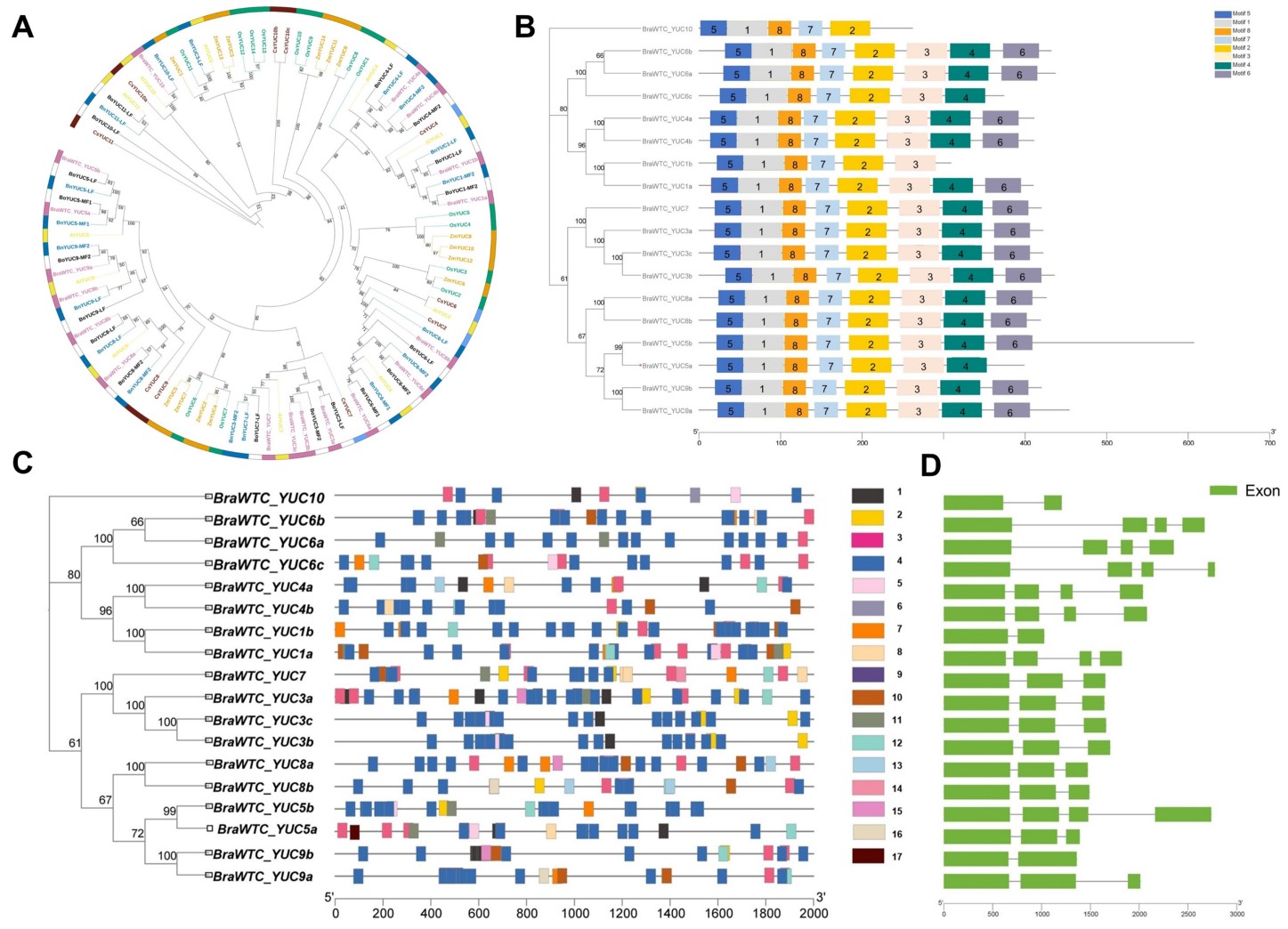

**Figure 1 Evolutionary tree, conservative motif, gene structure and prediction of *cis*-acting elements analysis of *YUC* gene family.** (A) Evolutionary tree of *YUC* gene family in 'Wutacai', *B. oleracea*, *B. nigra*, Arabidopsis, cucumber, rice, and maize. Different colors represent different species, and the numerical values in the figure represent bootstraps values. LF, Least Fractioned subgenome. MFs (MF1 and MF2), More fractioned subgenomes. (B) Conservative motif analysis of the *YUC* gene family. Different color boxes and numbers indicate different conserved motifs, and black lines indicate amino acid sequences. The left side shows the phylogenetic tree of the *YUC* gene family members in 'Wutacai'. And the numerical values in the evolutionary tree represent bootstraps values. (C) Prediction of *cis*-acting elements of *YUC* gene family. Different color boxes represent different types of *cis*-acting elements, and gray lines represent gene sequences. 1, Auxin response; 2, Defense and stress response; 3, Anaerobic induction; 4, Light response; 5, Meristem expression; 6, Palisade mesophyll cell differentiation; 7, Low temperature response; 8, Salicylic acid response; 9, Abscisic acid response; 10, Methyl jasmonate response; 11, Drought induction; 12, Gibberellin response; 13, Physiological control; 14, Endosperm expression; 15, Regulation of zein metabolism; 16, Gene regulation of flavonoid biosynthesis; 17, Down-regulated expression of photochrome. (D) Analysis of gene structure.

## Analyses of physicochemical properties, conserved motifs, and subcellular localization

The results of primary structure prediction unveiled that the *BraWTC_YUCs* encoded proteins of 262 to 607 amino acid residues in length, with corresponding molecular weights of 29.66 to 68.24 kDa. The isoelectric points fell within the range of 5.99 to 9.46, and the protein instability index ranged from 27.82 to 50.55 (Table 1). According to the established criteria, proteins with an instability index of less than 40 are considered stable,

**Table 1 Physicochemical properties of YUC family proteins of 'Wutacai'.**

| Gene | Gene ID | Number of amino acids | Molecular weight (kD) | Theoretical pI | Instability index | Grand average of hydropathicity (GRAVY) | Subcellular localization (Localization scores) |
|---|---|---|---|---|---|---|---|
| *BraWTC_YUC1a* | Ann023190.1_BraWTC | 410 | 45.23 | 9.46 | 39.17 | −0.151 | Cytoplasmic membrane |
| *BraWTC_YUC1b* | Ann404350.1_BraWTC | 309 | 34.16 | 9.46 | 39.26 | −0.052 | Cytoplasmic membrane |
| *BraWTC_YUC3a* | Ann445240.1_BraWTC | 422 | 47.11 | 9.02 | 35.91 | −0.201 | Cytoplasmic membrane |
| *BraWTC_YUC3b* | Ann370100.1_BraWTC | 436 | 48.65 | 9.04 | 36.85 | −0.238 | Cytoplasmic membrane |
| *BraWTC_YUC3c* | Ann315510.1_BraWTC | 422 | 47.00 | 8.82 | 35.22 | −0.199 | Cytoplasmic membrane |
| *BraWTC_YUC4a* | Ann246750.1_BraWTC | 411 | 45.62 | 9.35 | 34.72 | −0.055 | Cytoplasmic membrane |
| *BraWTC_YUC4b* | Ann040630.1_BraWTC | 411 | 45.58 | 9.21 | 33.92 | −0.079 | Cytoplasmic membrane |
| *BraWTC_YUC5a* | Ann125900.1_BraWTC | 399 | 44.27 | 8.76 | 45.89 | −0.125 | Cytoplasmic membrane |
| *BraWTC_YUC5b* | Ann300290.1_BraWTC | 607 | 68.24 | 6.48 | 49.90 | −0.231 | Cytoplasmic membrane |
| *BraWTC_YUC6a* | Ann173900.1_BraWTC | 437 | 48.68 | 8.92 | 39.64 | −0.160 | Cytoplasmic membrane |
| *BraWTC_YUC6b* | Ann075860.1_BraWTC | 432 | 48.36 | 8.99 | 42.79 | −0.216 | Nucleus |
| *BraWTC_YUC6c* | Ann469230.1_BraWTC | 374 | 41.77 | 8.67 | 33.30 | −0.197 | Cytoplasmic membrane |
| *BraWTC_YUC7* | Ann432590.1_BraWTC | 420 | 47.03 | 8.87 | 38.95 | −0.196 | Cytoplasmic membrane |
| *BraWTC_YUC8a* | Ann021460.1_BraWTC | 426 | 47.84 | 8.87 | 47.39 | −0.208 | Cytoplasmic membrane |
| *BraWTC_YUC8b* | Ann407190.1_BraWTC | 419 | 46.95 | 8.87 | 49.78 | −0.208 | Cytoplasmic membrane |
| *BraWTC_YUC9a* | Ann315300.1_BraWTC | 454 | 50.46 | 9.07 | 49.11 | −0.152 | Cytoplasmic membrane |
| *BraWTC_YUC9b* | Ann444810.1_BraWTC | 420 | 46.88 | 9.37 | 50.55 | −0.165 | Cytoplasmic membrane |
| *BraWTC_YUC10* | Ann146080.1_BraWTC | 262 | 29.66 | 5.99 | 27.82 | −0.324 | Cytoplasmic membrane |

while those with an isoelectric point below seven are categorized as acidic, and proteins with a hydrophilicity index less than zero are classified as hydrophilic. Therefore, it can be inferred that BraWTC_YUC5a/b, BraWTC_YUC6b, BraWTC_YUC8a/b, and BraWTC_YUC9a/b were stable proteins. BraWTC_YUC5b and BraWTC_YUC10 were acidic proteins, while the remaining members were classified as alkaline proteins. All 18 members were identified as hydrophilic proteins. Analysis of conserved motifs revealed that BraWTC_YUCs proteins harbored five to eight conserved motifs, including motif 1, motif 2, motif 5, motif 7, and motif 8. With the exception of BraWTC_YUC10, the other members exhibited motif 3, and motif 4 was shared by the remaining members, excluding BraWTC_YUC10 and BraWTC_YUC1b. Additionally, except for BraWTC_YUC1b/5a/6c/10, motif 6 was present in all other YUC gene family members (Fig. 1B). This observation suggests a certain level of conservation in the YUC protein motifs within 'Wutacai'. Subcellular localization analysis, *via* bioinformatic analysis, inferred that BraWTC_YUC6b was localized in the nucleus, whereas the remaining members were localized in the cytoplasmic membrane (Table 1).

## Analysis of the gene structure and prediction of *cis*-regulatory elements

The 2,000 bp upstream sequences of *BraWTC_YUCs* were extracted to serve as promoter regions for the identification of *cis*-regulatory elements. The outcomes revealed that these

sequences encompassed diverse types of *cis*-regulatory elements, primarily categorized into three groups: plant hormone-related elements, growth and development-related elements, and stress response-related elements (Fig. 1C). The plant hormone-related elements comprised response elements for auxin, salicylic acid, abscisic acid, methyl jasmonate, and gibberellin. Meanwhile, the growth and development-related elements such as light response, meristem expression, epidermal cell differentiation, physiological control, endosperm expression. Lastly, the stress response-related elements encompassed defense and stress response, anaerobic induction, low-temperature response, and drought induction. The types of *cis*-regulatory elements within *BraWTC_YUCs* exhibited variability. Specifically, *BraWTC_YUC6a* displayed the fewest response elements, comprising only four element types, whereas *BraWTC_YUC8b* featured the most response elements, encompassing ten element types. Within the plant hormone response elements, *BraWTC_YUC10* and *BraWTC_YUC5b* each possessed only one type of plant hormone response element, specifically auxin response and gibberellin response, respectively. *BraWTC_YUC5a* contained four response elements excluding methyl jasmonate, and *BraWTC_YUC8b* featured four response elements excluding auxin. All members were found to contain a substantial number of light response elements, with *BraWTC_YUC8b* featuring as many as 22 element types of light response elements, while *BraWTC_YUC10* exhibited the fewest (four element types). Regulatory elements for meristem expression were present in *BraWTC_YUC1a*, *BraWTC_YUC3b/c*, *BraWTC_YUC5a/b*, *BraWTC_YUC6c*, and *BraWTC_YUC10*. Only a select few members featured elements related to flavonoid biosynthesis gene regulation, physiological control, and zein protein (major storage protein of corn) metabolism regulation. Specifically, *BraWTC_YUC7* and *BraWTC_YUC10* contained response elements for epidermal cell differentiation and endosperm expression, respectively, which were absent in other members. Gene structure analysis showed that *BraWTC_YUCs* had two to four exons (Fig. 1D). *BraWTC_YUC1b*, *BraWTC_YUC9b*, and *BraWTC_YUC10* owned two exons, while there are four exons in *BraWTC_YUC7*, *BraWTC_YUC3a/b/c*, *BraWTC_YUC8a/b*, *BraWTC_YUC5a* and *BraWTC_YUC9a*.

## Expression of *BraWTC_YUCs* in Chinese cabbage exhibits organ-specific and leaf position-dependent variation

Based on the expression data of *YUCs* in Chinese cabbage extracted from the BRAD database, it is evident that the expression profiles of the *YUCs* vary significantly across different organs. Notably, in roots, *BraWTC_YUC5*, *BraWTC_YUC6* and *BraWTC_YUC8* showed higher expression levels, while in stems, *BraWTC_YUC4* and *BraWTC_YUC6* exhibited dominant expression. In flowers and siliques, *BraWTC_YUC6*, *BraWTC_YUC8* and *BraWTC_YUC10* displayed higher expression levels compared to other members. Expression levels were relatively low in callus tissue, whereas in leaves, overall expression levels of *YUCs* at the sampling period (7 weeks after sowing) were relatively low. However, *BraWTC_YUC6* showed stable expression in leaves compared to the expression level of other family members. Additionally, distinct copies of the same gene, such as *BraWTC_YUC3/4/5/8*, exhibited expression pattern variations. Specifically,

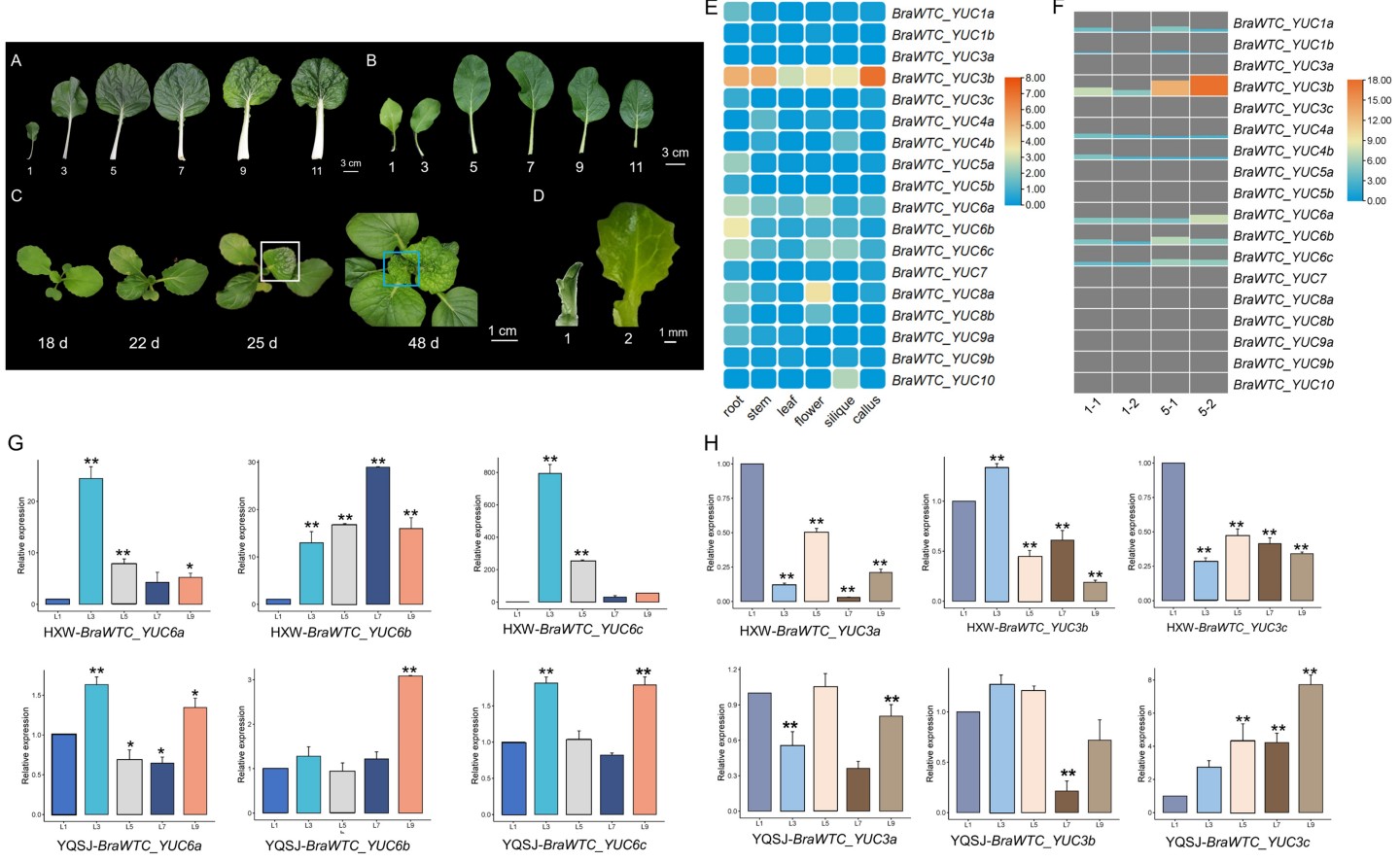

**Figure 2 Leaf development of Chinese cabbage and the expression profile of *BraWTC_YUCs*.** (A) 1st, 3rd, 5th, 7th, 9th and 11th mature leaves of 'Wutacai' at 71 Days after sowing (DAS). (B) 1st, 3rd, 5th, 7th, 9th and 11th mature leaves of 'Youqing 49' at 35 DAS. (C) 'Wutacai' on 18, 22, 25, and 48 DAS, the white boxes showed local wrinkling in the 4th true leaf at juvenile stage. The blue boxes showed total wrinkling in the 9th true leaf at juvenile stage, respectively. (D) Two periods of sampling for transcriptome sequencing. (E) Expression of *YUC* gene in different organs of Chinese cabbage (data from BRAD Database). The value in the figure is the TPM value processed by taking the logarithm to the base 2 (Transcripts Per Kilobase of exon model per Million mapped reads). (F) The expression of *YUC* gene at 1st and 5th leaf of 'Wutacai' in transcriptome sequencing results. 1-1 and 1-2 represent two periods at 1st leaf, and 5-1 and 5-2 represent two periods at 5th leaf, respectively. Value in the figure represents the FPKM value (Fragments Per Kilobase of exon model per Million mapped fragments). (G) RT-qPCR analysis of *BraWTC_YUC6a/b/c* expression in the 1st, 3rd, 5th, 7th and 9th mature leaves of 'Youqing 49' and 'Wutacai'; L1/3/5/7/9 represents the 1st, 3rd, 5th, 7th, 9th mature leaves respectively; YQSJ represents 'Youqing 49', HXW represents 'Wutacai'. (H) RT-qPCR analysis of *BraWTC_YUC3a/b/c* expression level in the 1st, 3rd, 5th, 7th and 9th mature leaves of 'Youqing 49' and 'Wutacai' and $^*p < 0.05$ compared to L1; $^{**}p < 0.01$ compared to L1.   

*BraWTC_YUC3a* was not expressed in leaves, *BraWTC_YUC3c* showed low expression, while *BraWTC_YUC3b* exhibited relatively high expression, being the highest among all detected *YUCs*. Additionally, *BraWTC_YUC1*, *BraWTC_YUC7* and *BraWTC_YUC10* showed no expression in leaves (Fig. 2E).

According to the expression data, *BraWTC_YUC3* and *BraWTC_YUC6* showed higher expression levels in the leaves compared to other members. Therefore, RNA was extracted from mature leaves (1st, 3rd, 5th, 7th, 9th leaves) of the flowering Chinese cabbage 'Youqing 49' and 'Wutacai' to further analyze the expression levels of *BraWTC_YUC3* and *BraWTC_YUC6*. Our preliminary observations revealed that the leaf surfaces of the 1st to 3rd true leaves of 'Wutacai' were flat, resembling flat-leaved varieties such as 'Suzhouqing'

and 'Youqing 49' (Fig. 2B), without any wrinkling. However, the 4th true leaf exhibited localized wrinkling during the heart leaf stage, which either became flat or retained localized wrinkling upon maturation. Leaves from the 5th to 8th positions showed localized wrinkling during the juvenile stage, which persisted after maturation. Starting from the 9th leaf, the leaves exhibited a higher degree of wrinkling, irrespective of their maturity (Figs. 2A and 2C). RT-qPCR results showed variations in the expression levels of *BraWTC_YUC3* and *BraWTC_YUC6* among different leaf positions, with a more pronounced difference observed in 'Wutacai' (Figs. 2G and 2H). Additionally, the expression difference of *BraWTC_YUC6* between 'Youqing 49' and 'Wutacai' was more significant compared to *BraWTC_YUC3*.

## Transcriptome analysis reveals the potential involvement of *YUC* genes in the leaf wrinkling trait of Chinese cabbage

In-depth investigation was carried out to explore the involvement of YUCs in the development of leaf wrinkling in 'Wutacai', particularly during the early leaf developmental stages. Transcriptome sequencing was employed to analyze the expression profiles of YUCs in the 1st and 5th leaves of 'Wutacai' and 'Suzhouqing' during the juvenile stages (Fig. 2D). Differentially expressed genes (DEGs) were identified through comparisons among different groups: H1-1 *vs* S1-1, H1-2 *vs* S1-2, H5-1 *vs* S5-1, and H5-2 *vs* S5-2. The analysis revealed 5894 DEGs in H1-1 *vs* S1-1, 6304 DEGs in H1-2 *vs* S1-2, 5211 DEGs in H5-1 *vs* S5-1, and 8446 DEGs in H5-2 *vs* S5-2.

The GO enrichment analysis for DEGs in these groups highlighted pathways predominantly associated with "biological process", "integral component of membrane" and "ATP binding". KEGG enrichment analysis was conducted on the DEGs across the four comparison groups, revealing that the top 20 enriched pathways encompassed five main categories: metabolism, genetic information processing, environmental information processing, cellular processes, and organismal systems. Notably, DEGs in all four comparison groups consistently enriched pathways in the cellular processes and organismal systems categories, particularly associated with plant hormone signal transduction. Moreover, both the H1-1 *vs* S1-1 and H5-1 *vs* S5-1 comparison groups exhibited enrichment in the auxin biosynthesis pathway and tryptophan metabolism (Fig. 3).

Upon analyzing the enrichment results of DEGs associated with the pathways "Plant hormone signal transduction" and "Tryptophan metabolism" combined with existing literature pointing towards the influence of auxin on the development of leaf wrinkling in Chinese cabbage, a comprehensive examination of the pertinent DEGs involved in auxin synthesis, metabolism, and signaling pathways was conducted. The findings unveiled distinct expression patterns of key regulatory enzymes in auxin biosynthesis, particularly members of the *YUC* gene family, across all comparison groups. Notably, *BrYUC6* and *BrYUC1* exhibited significantly reduced expression levels at the 1st and 5th leaf positions in and 'Suzhouqing' compared to those in 'Wutacai'. Additionally, variations in the expression of genes associated with auxin transport and downstream responsive genes within the signaling pathway were also observed (Fig. 4).

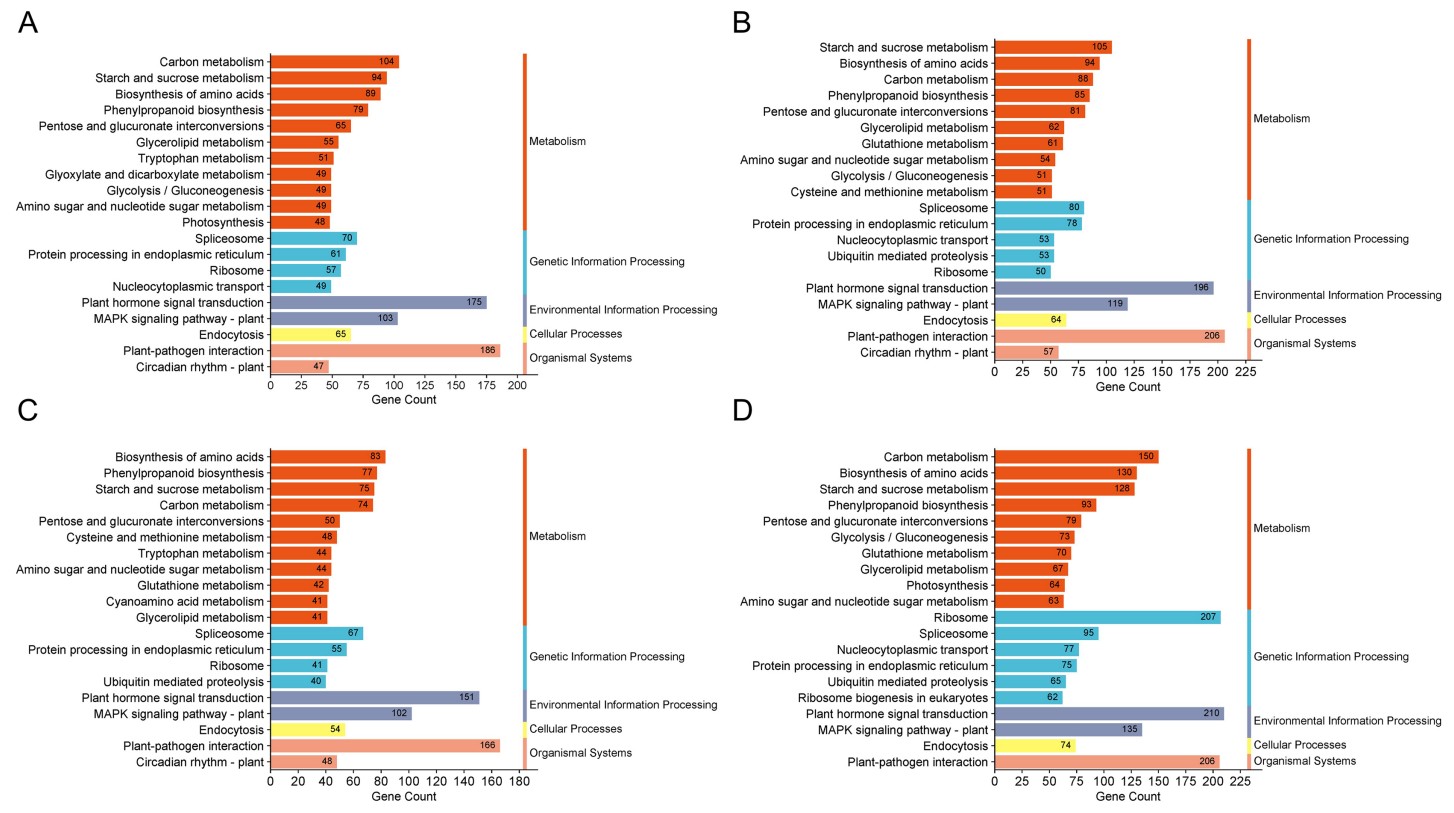

**Figure 3 KEGG enrichment column of differential expressed genes.** (A) H1-1 *vs* S1-1. (B) H1-2 *vs* S1-2. (C) H5-1 *vs* S5-1. (D) H5-2 *vs* S5-2.

## DISCUSSION

Auxin plays an important regulatory role in establishing polarity and flattening during leaf development (*Merelo et al., 2017*; *Wang et al., 2022a*). Transcriptome sequencing analysis conducted on the wrinkled and flat parts of mature leaves in 'Wutacai' revealed that the differentially expressed genes between these regions were predominantly enriched in the auxin signaling pathway (*Hou et al., 2023*), indicating the involvement of auxin in the formation or maintenance of leaf wrinkling trait in Chinese cabbage. In this study, the content of endogenous hormones in two different forms (flat and wrinkled) was measured. The results indicated differential levels of six types of auxins, which might contribute to the leaf wrinkling of Tacai. However, the study mainly focused on the role of auxin signaling transduction in leaf development, leaving the relationship between local auxin synthesis and leaf wrinkling unexplored.

In recent years, an increasing number of studies have shown that local synthesis of auxin is closely related to leaf morphogenesis. The TAA/YUC pathway is the main endogenous biosynthetic pathway for auxin (*Gomes & Scortecci, 2021*). Leaf primordia originate from the differentiation of meristematic tissues, and leaf polarity is established through proximal-distal, medio-lateral, and adaxial-abaxial axes, which then elongate to form leaves (*Byrne, 2012*). The establishment of proximal-distal polarity involves localized auxin

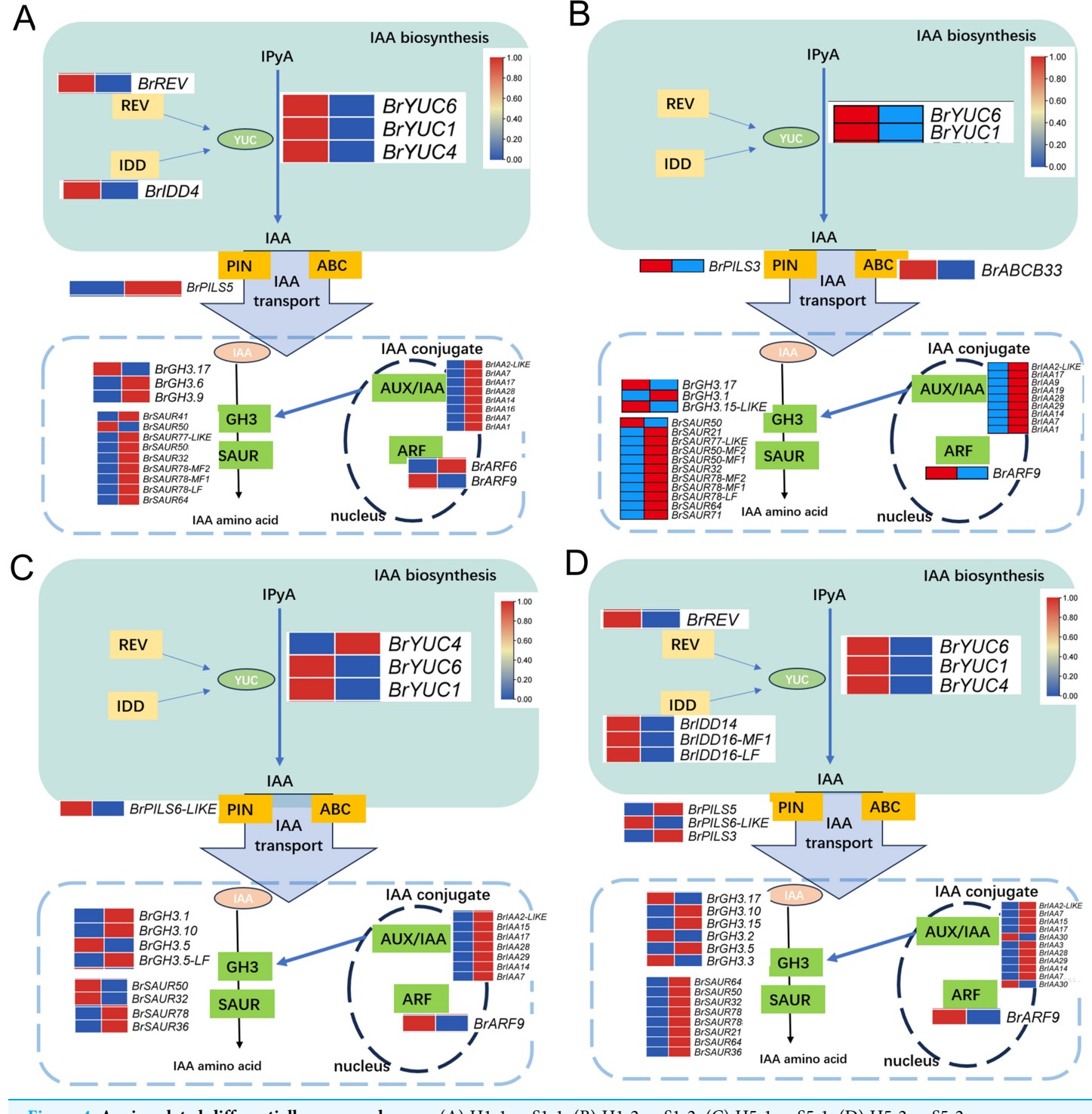

**Figure 4 Auxin related differentially expressed genes.** (A) H1-1 *vs* S1-1. (B) H1-2 *vs* S1-2. (C) H5-1 *vs* S5-1. (D) H5-2 *vs* S5-2.

accumulation in leaf margin cells mediated by multiple YUCs (*Zgurski et al., 2005*; *Wang et al., 2011*). Studies have indicated that the elimination of multiple *YUCs* leads to narrow leaf phenotypes due to auxin deficiency (*Cheng, Dai & Zhao, 2007*), while excessive auxin

leads to leaf curling, indicating the involvement of local auxin concentration in the formation of flat leaf surfaces. Based on this, it can be speculated that the *YUC* gene family may be involved in the formation of leaf wrinkling trait. Therefore, conducting a comprehensive investigation of the *YUC* gene family in Chinese cabbage is necessary to reveal the genetic regulatory mechanisms of leaf wrinkling trait.

Phylogenetic analysis of the *YUC* gene family in 'Wutacai', *B. oleracea*, *B. nigra*, Arabidopsis, cucumber, rice, and maize revealed relatively distant evolutionary relationships as well as between 'Wutacai' and rice. This distinction may be attributed to the monocotyledonous nature of maize and rice, while 'Wutacai' is a dicotyledonous plant. *YUCs* in 'Wutacai' (AA, 2n = 20) demonstrated a close evolutionary relationship with those of *B. oleracea* (CC, 2n = 18) and *B. nigra* (BB, 2n = 16), likely due to the inherent relationships within the *Brassica* genus. In the 'Wutacai' genome, a total of 18 *YUCs* were identified, indicating the expansion of the *YUC* gene family in 'Wutacai' concurrent with whole genome duplication events. This phenomenon also happened in *B. oleracea* and *B. nigra*. Compared to Arabidopsis, the *Brassica* genus experienced a whole genome triplication event around 5.4 to 9 million years ago (*Cheng, Wu & Wang, 2014*).

*YUCs* encode a flavin-containing monooxygenase, FMO, which was first discovered in plants. They exhibit a typical structural and functional characteristic similar to mammalian FMOs, typically comprising six highly conserved motifs (*Schlaich, 2007*). Protein structure analysis revealed that BraWTC_YUCs contain 5–8 conserved motifs, all of which encompass FAD and NADPH binding motifs. Most of these proteins exhibited alkaline properties and displayed hydrophilic characteristics. Subcellular localization results indicated that, apart from BraWTC_YUC6b, which localized to the nucleus, others were localized to cytoplasmic membrane. Promoter analysis predicted the presence of various *cis*-acting elements associated with plant hormones, growth, and development, as well as stress responses in the promoter regions of 'Wutacai' *YUCs*. This suggests that *YUCs* may be involved in a broad spectrum of biological activities.

In Arabidopsis, the biological functions of the *YUCs* have been extensively studied, including their involvement in root development, leaf development, reproductive development, and response to abiotic stress (*Cao et al., 2019*). However, a previous study identifying the *YUC* gene family in Chinese cabbage (*Qi et al., 2019*) did not explore their expression and function. Therefore, investigating the function of *YUCs* in Chinese cabbage represents an intriguing avenue for research.

Intriguingly, our study unveiled, for the first time, an association between the leaf wrinkling trait and leaf position in 'Wutacai'. Analogous circumstances arise in the head formation of Chinese cabbage, with variations in leaf characteristics observed across different leaf positions. Studies indicate that the key transition leaf in the heading process of Chinese cabbage is the 7th true leaf, regulated by an intricate signaling network (*Guo et al., 2022*). Hence, we utilized leaf samples from the 1st and 5th leaves, which exhibited significant differences during both the juvenile and mature stages, to establish a comprehensive expression profile of *BrYUCs* during leaf wrinkling formation in Chinese cabbage. In accordance with previous studies, the occurrence of curly leaves is associated with failure or damage to leaf polarity establishment during leaf primordium development

(*Liu et al., 2011*). Thus, gene regulation takes place earlier than the completion of leaf morphogenesis during leaf development. Focusing on early leaf morphogenesis rather than the mature leaf stages for transcriptome sequencing analysis of two cultivars with distinct leaf features, it allowed us to obtain a better understand the relationship of *YUCs* to leaf wrinkling, which leads to the identification of several candidate genes such as *BrYUC1* and *BrYUC6*. As anticipated, the expression levels of *BraWTC_YUCs* varied at different time points within the same leaf position, indicating dynamic changes in *YUC* gene expression during leaf development. Therefore, we propose that the *YUC* gene family is implicated in the intricate biological process of the leaf wrinkling trait in Chinese cabbage. Nevertheless, further investigations are required to elucidate the specific roles they play in this process.

## CONCLUSION

In this study, we focused on the distinctive cultivar 'Wutacai' of the Tacai variety, renowned for its prominent leaf wrinkling trait. A total of 18 *YUCs* were identified in the Chinese cabbage genome. Protein structure and evolutionary relationship analysis revealed the conservation of *YUCs* within the genus *Brassica*. Through the transcriptome sequencing analysis of two cultivars with distinct leaf features, we established a comprehensive expression profile of *BrYUCs* and unveiled the potential involvement of *YUCs* in leaf wrinkling formation. These findings serve as a theoretical reference for further exploration of the regulatory mechanisms underlying the leaf wrinkling phenotype in Chinese cabbage and other plant species which also display a wrinkled leaf phenotype.

### Funding

This work was supported by the National Natural Science Foundation of China (32372728), the Key R&D Program of Zhejiang (2022C02032 & 2022C02030), the Grand Science and Technology Special Project of Zhejiang Province (2021C02065), the SanNongJiuFang Science and Technology Cooperation Project of Zhejiang Province (2023SNJF008), and the Hainan Provincial Joint Project of Sanya Yazhou Bay Science and Technology City (2021JJLH0030). There was no additional external funding received for this study. The funders had no role in study design, data collection and analysis, decision to publish, or preparation of the manuscript.

### Grant Disclosures

The following grant information was disclosed by the authors:
National Natural Science Foundation of China: 32372728.
Key R & D Program of Zhejiang: 2022C02032 & 2022C02030.
Grand Science and Technology Special Project of Zhejiang Province: 2021C02065.
SanNongJiuFang Science and Technology Cooperation Project of Zhejiang Province: 2023SNJF008.
Hainan Provincial Joint Project of Sanya Yazhou Bay Science and Technology City: 2021JJLH0030.

## Competing Interests

The authors declare that they have no competing interests.

## Author Contributions

- Xuelian Ye conceived and designed the experiments, performed the experiments, analyzed the data, prepared figures and/or tables, authored or reviewed drafts of the article, and approved the final draft.
- Ji Sun performed the experiments, prepared figures and/or tables, and approved the final draft.
- Yuan Tian conceived and designed the experiments, performed the experiments, analyzed the data, authored or reviewed drafts of the article, and approved the final draft.
- Jingwen Chen performed the experiments, analyzed the data, authored or reviewed drafts of the article, and approved the final draft.
- Xiangtan Yao analyzed the data, prepared figures and/or tables, and approved the final draft.
- Xinhua Quan performed the experiments, prepared figures and/or tables, authored or reviewed drafts of the article, and approved the final draft.
- Li Huang conceived and designed the experiments, authored or reviewed drafts of the article, and approved the final draft.

## DNA Deposition

The following information was supplied regarding the deposition of DNA sequences:

The raw RNA-seq data of the transcriptome is available at Genome Sequence Archive (Genomics, Proteomics & Bioinformatics 2021) in National Genomics Data Center (Nucleic Acids Res 2022), Beijing Institute of Genomics, Chinese Academy of Sciences: CRA015400 and CRA015503.

## Data Availability

The raw measurements are available in the Supplemental Files.

## Supplemental Information

Supplemental information for this article can be found online at http://dx.doi.org/10.7717/peerj.17337#supplemental-information.

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
