# Peer review of "Identification of YUC genes associated with leaf wrinkling trait in Tacai variety of Chinese cabbage"

_PeerJ, doi:10.7717/peerj.17337_

## Round 0.1 · original submission · Major Revisions

Dear authors,

Your manuscript has now been reviewed by two experts in the field, and both reviewers have made the same recommendation of 'Major Revision'. My decision is therefore in line with that of Reviewer #1 and #2 of 'Major Revision'.

Please carefully consider the concerns raised by the two reviewers, especially those raised by Reviewer #2 in their more detailed assessment of your study. Each of the concerns raised by the two reviewers must be appropriately addressed as part of authoring a significantly improved, and revised version of your manuscript.

Please also provide a detailed 'Letter of Response' that accompanies your revised manuscript so that I can determine whether the required improvements have been made to the revised manuscript version.

Figure quality and a lack of statistical analysis also forms a concern regarding the experimental rigor of your study. As part of the revision process, please improve the quality of the manuscript Figures, and conduct the appropriate statistical analysis for thorough interpretation of your experimental findings.

Please take your time in generating the revised manuscript version as a lot of work is required to improve the standard of your manuscript so that it can be considered further in the PeerJ system.

Kind regards,
Andy Eamens

**Language Note:** PeerJ staff have identified that the English language needs to be improved. When you prepare your next revision, please either (i) have a colleague who is proficient in English and familiar with the subject matter review your manuscript, or (ii) contact a professional editing service to review your manuscript. PeerJ can provide language editing services - you can contact us at [email protected] for pricing (be sure to provide your manuscript number and title). – PeerJ Staff

Reviewer 1 ·

Basic reporting

No comments.

Experimental design

We can not find the relationship betweenYUC genes and wrankling traints. The author should explain why YUC genes are associated with wrankling traints in Brassica campestris ssp. chinensis. So it is better if the authors could carry out an experiment to confirm it.

Validity of the findings

.

Additional comments

This manuscript title is "Identiûcation of YUC genes associated with leaf wrinkling
traits in Brassica campestris ssp. chinensis var. rosularis".
Could you please tell us if YUC genes would be associated with leaf wrinkling traits in Brassica campestris ssp. chinensis?

Reviewer 2 ·

Basic reporting

The YUC gene family from ‘Wutacai’ was identified in this study, and the expression patterns of YUC gene family members in flat and wrinkled leaves were analyzed.

Experimental design

Many bioinformatics analyses were employed, but experiment results were lack, which reduced the readability of the article.

Validity of the findings

1. In Introduction and Discussion, the authors should add more evidences or references about that the local concentration of auxin was related to the leaf wrinkling phenotype. Determination or analysis of the local concentration of auxin at different leaf positions will contribute to elucidate the possible roles of YUC gene in leaf development. Please increase the resolution of Figure 5B and 5C.

2. The expression patterns of YUC genes in different developmental stages of leaves were performed in ‘Wutacai’ and the flat-leaved representative variety ‘Youqing 49’. The presentation of leaf characteristics in the two varieties should be exhibited in Figure 5 or others.

3. The phylogenetic tree of YUC genes was analyzed to explore the evolution of homologous genes. The homologous genes from other Brassica crops should be selected, instead of the monocotyledonous plants, maize and rice.

4. The data of transcriptome sequencing in ‘Wutacai’ for expression profiles of YUC gene family should be presented or added the related public database or references.

5. The written needed to be improved using Language Polishing.

6. The data in this study lack Statistic analysis.

7. In ‘Arabidopsis’, the initial needed to be capitalized. Line 128, 139 and others.

8. Line 103: please added the correct reference format.

---

## Round 0.2 · Minor Revisions

Dear authors,

I have reviewed the revised version of your manuscript myself and have identified numerous, yet small, text-related issues that required further attention.

Please use the annotated PDF as a reference to correct all outstanding issues that remain with the revised manuscript version of your study.

Once these changes (please make all requested changes) have been made/addressed by the authorship team, please resubmit your further revised manuscript for further consideration.

Please also include a 'Letter of Response' which documents each change made to this version of your manuscript when submitting your revised manuscript.

Kind regards,
Andrew Eamens

---

## Round 0.3 · accepted · Accept

Dear authors,

I am of the opinion that all comments/concerns that I raised in my review of the previous version of this study have been adequately addressed in the revised manuscript (Version #2).

I did not send the revised manuscript back out to the original peer reviewers as I reviewed the manuscript myself to determine whether the authorship team had made the changes that I had requested in my previous review of the manuscript.

The manuscript is now of a standard that is acceptable for publication. Please do carefully follow all listed tasks assigned to you as part of the subsequent publication process.

Kind regards,
Andrew Eamens